# Abnormal Brain Structure Is Associated with Social and Communication Deficits in Children with Autism Spectrum Disorder: A Voxel-Based Morphometry Analysis

**DOI:** 10.3390/brainsci13050779

**Published:** 2023-05-10

**Authors:** Ming-Xiang Xu, Xing-Da Ju

**Affiliations:** 1School of Psychology, Northeast Normal University, Changchun 130024, China; xmx19970510@163.com; 2Jilin Provincial Key Laboratory of Cognitive Neuroscience and Brain Development, Changchun 130024, China

**Keywords:** autism spectrum disorder, magnetic resonance imaging, voxel-based morphometry, gray matter volume

## Abstract

Structural magnetic resonance imaging (sMRI) studies have shown abnormalities in the brain structure of ASD patients, but the relationship between structural changes and social communication problems is still unclear. This study aims to explore the structural mechanisms of clinical dysfunction in the brain of ASD children through voxel-based morphometry (VBM). After screening T1 structural images from the Autism Brain Imaging Data Exchange (ABIDE) database, 98 children aged 8–12 years old with ASD were matched with 105 children aged 8–12 years old with typical development (TD). Firstly, this study compared the differences in gray matter volume (GMV) between the two groups. Then, this study evaluated the relationship between GMV and the subtotal score of communications and social interaction on the Autism Diagnostic Observation Schedule (ADOS) in ASD children. Research has found that abnormal brain structures in ASD include the midbrain, pontine, bilateral hippocampus, left parahippocampal gyrus, left superior temporal gyrus, left temporal pole, left middle temporal gyrus and left superior occipital gyrus. In addition, in ASD children, the subtotal score of communications and social interaction on the ADOS were only significantly positively correlated with GMV in the left hippocampus, left superior temporal gyrus and left middle temporal gyrus. In summary, the gray matter structure of ASD children is abnormal, and different clinical dysfunction in ASD children is related to structural abnormalities in specific regions.

## 1. Introduction

Autism spectrum disorder (ASD) is a lifelong neurodevelopmental disorder, and its characteristics are a lack of communication and social interaction, as well as symptoms of stereotyped repetitive behaviors and restricted interests [1,2]. The prevalence of ASD is increasing year by year. According to the Centers for Disease Control and Prevention of the United States, the prevalence of ASD is up to 1/36, seriously damaging the social function of patients [3]. In recent years, ASD research based on structural magnetic resonance image (sMRI) technology has attracted the interest of many researchers. Numerous studies based on brain MRI data have reported specific morphological changes in ASD patients compared to typically developing (TD) individuals, such as brain area volume and density, cortical thickness, cortical surface area, and local fold index [4]. Previous studies have shown that the etiology of ASD may be due to the interaction between genetic and environmental factors, but its pathological mechanism is still unclear [5,6]. Although the etiology of ASD has not been determined, abnormal brain structure plays an important role in the development of ASD. Compared to normally developing individuals, changes in brain structure often occur in ASD patients [7,8,9].

The prefrontal cortex has been proven to be involved in cognitive processing, emotional understanding, social behavior and working memory [10,11,12]. Research showed that a portion of ASD patients have an increase in frontal lobe gray matter volume (GMV) [13]. However, Abell [14] found that compared with TD individuals, ASD patients showed reduced GMV in the left inferior frontal gyrus, while they showed increased GMV in the amygdala, middle temporal gyrus, inferior temporal gyrus and cerebellum. Another study also showed reduced GMV in the left frontal lobe and dorsolateral prefrontal cortex in adolescents with ASD [15]. In addition, the temporal lobe structure of ASD patients is considered abnormal, and their temporal lobe structures, such as the right temporal pole and right inferior temporal gyrus, have a larger GMV than the TD group [16]. Unusually destructive behaviors in ASD may be related to damage to the amygdala and related temporal lobe structures. The brain areas responsible for the social function of ASD mainly include the amygdala, hippocampus and superior temporal gyrus [17]. Some studies found that ASD patients showed increased GMV in the prefrontal lobe, middle temporal gyrus and superior temporal gyrus [18,19]. However, contradictory conclusions have been drawn from an earlier study that no abnormalities in temporal GMV were found in ASD patients when gender and age were controlled [20]. Therefore, structural changes in gray matter in ASD patients still need to be further explored with a larger sample size.

In fact, detecting changes in brain structure is crucial for revealing the pathological mechanisms of ASD. In particular, these brain regions with significant differences can be identified as neuroimaging biomarkers associated with the disease [21]. One study has identified biomarkers for neuroimaging, such as the gyrus rectus, left middle temporal gyrus, inferior temporal gyrus and left inferior frontal gyrus [22]. Further research has found that the communication skill score on the Autism Diagnostic Observation Schedule (ADOS) was negatively correlated with the regional GMV of the gyrus rectus, left middle temporal gyrus, and inferior temporal gyrus; there is a significant negative correlation between communication skill score on ADOS and the orbital part of the left inferior frontal gyrus. Moreover, GMV reduction was also detected in the left inferior and superior frontal gyrus, hippocampi and cerebellum by comparing 26 ASD children with 21 controls, and the resulting altered regions correlated with the Autism Diagnostic Interview Revised (ADI-R) and ADOS [23]. The reason for involving numerous brain regions is that ASD is a complex neurodevelopmental disorder, and the quantitative meta-analysis indicated that ASD is unlikely to be associated with anomalies in a specific region alone [24].

Most studies have found abnormal brain structures in ASD patients, but the relationship between structural changes and social communication problems is still unclear. Mitchell [25] found a reduction in prefrontal lobe GMV in the ASD group, which is associated with social and communication scores on the ADOS. However, up to now, there is still a lack of research on the relationship between the gray matter structure of ASD children and social interaction and communication dysfunction. Meanwhile, the brain development and behavioral development of ASD patients are closely related to the intelligence quotient (IQ) of participants. Considering factors such as gender, age, and IQ is crucial for understanding the neural mechanisms of brain and behavioral development in ASD. Therefore, this study used a large number of subjects obtained from the Autism Brain Imaging Data Exchange (ABIDE) database in order to investigate the relationship between gray matter structure abnormalities and clinical symptoms in ASD children. 

## 2. Materials and Methods

### 2.1. Participants and MRI Data Collection

ABIDE is a publicly available structural MRI and resting-state fMRI database that collected data from 17 independent sites for ASD patients and TD patients, as described in detail by Di Martino et al. [26]. ABIDE includes 1112 datasets, comprising 539 ASD patients and 573 TD patients. The original studies included in ABIDE have been approved by the Institutional Review Board (IRB) at each site. Most sites used ADI-R or ADOS for autism diagnosis. In addition to the relevant information on the diagnostic scale, each site also provides basic demography information about each subject, including age and gender. Most sites also include measures of intelligence level. Details of acquisition, informed consent, and site-specific protocols are available at http://fcon_1000.projects.nitrc.org/indi/abide/ accessed on 5 January 2023.

All data included in this study came from the ABIDE database. All T1-weighted MR images were collected from a 3T MRI scanner with a resolution of 1 × 1 × 1 mm^3^. The original studies included in ABIDE were approved by the local ethics committees. The ASD group was screened according to the following criteria: (1) only include children aged 8–12 in the study; (2) remove subjects with incomplete cortical coverage; (3) exclude subjects lacking Wechsler Abbreviated Scales of Intelligence (WASI) and ADOS, full IQ ≥ 80; (4) remove head movement greater than 2 mm and rotation greater than 2°; (5) exclude centers with fewer than 10 people in a single center. The TD group was screened according to the following criteria: (1) age, IQ and sex, matched with the ASD group; (2) remove subjects with incomplete cortical coverage; (3) remove head movement greater than 2 mm and rotation greater than 2°; (4) exclude centers with fewer than 10 people in a single center. The IQ of all subjects was measured by WASI, which contains the full intelligence quotient (FIQ), verbal intelligence quotient (VIQ) and performance intelligence quotient (PIQ) indicators. Clinical symptoms of the ASD patients were assessed by ADOS. Finally, a total of 98 ASD children and 105 TD children were included in the subsequent data analysis.

### 2.2. MRI Data Processing

All MRI data were processed using MATLAB R2021a software. Image preprocessing was conducted based on the CAT12 toolbox (http://dbm.neuro.uni-jena.de/cat12/ accessed on 1 March 2023). Voxel-based morphometry (VBM) analysis based on the CAT12 toolbox was performed for each participant. Firstly, we manually repositioned the T1-weighted structural images to ensure that the origin was close to the front connection. Secondly, T1-weighted images were registered to the Montreal Neurological Institute (MNI) spatial template, and anatomical segmentation of T1-weighted structural images using the segmentation template of ‘New Segment’ to extract the original image and volumes of GM, white matter (WM), and cerebrospinal fluid (CSF). Finally, Gauss smoothing of standardized images was performed (full width at half maximum (FWHM) = 8 mm) to improve the signal-to-noise ratio (SNR) of the MR images.

### 2.3. Statistical Analysis

This study conducted two-sample *t*-tests to compare the differences in GMV between the ASD and TD groups with age, sex, IQ and total intracranial volume as covariates. A VBM analysis was performed in the SPM12 software, and the smoothed gray matter images of the ASD and TD groups were tested by voxel-based two-sample *t*-tests. To reduce false positive errors in the statistical analysis, Gaussian random field correction was adopted in this study for multiple comparison correction. The threshold of the voxel level was set to *p* < 0.001, and the voxel number of clusters was considered significant if it was more than 100.

Using the subtotal score of communications and social interaction on ADOS as the dependent variable, GMV with significant differences as the independent variable, age, gender, intelligence and total intracranial volume as covariates, a linear regression model was used to analyze the relationship between GMV and clinical symptoms in the ASD group.

## 3. Results 

### 3.1. Demographic Data 

In this study, age, sex, and IQ were matched between the ASD and TD groups (*p* > 0.05), as shown in Table 1. 

### 3.2. MRI Results 

This study found that abnormal brain structures in ASD include the midbrain, pontine, bilateral hippocampus, left parahippocampal gyrus, left superior temporal gyrus, left temporal pole, left middle temporal gyrus and left superior occipital gyrus (Figure 1, Figure 2 and Figure 3 and Table 2). The GMV of these abnormal brain structures mentioned above has increased in all three clusters compared with the TD group.

A regression analysis only showed that the subtotal score of communications and social interaction on ADOS were significantly positively correlated with GMV in the left hippocampus, left superior temporal gyrus and left middle temporal gyrus.

## 4. Discussion

This study included 98 ASD children and 105 TD children, who were matched for age, gender, and IQ. Through VBM analyses, different brain regions showed significant differences in GMV between the ASD group and the TD group. Compared with the TD group, the ASD group showed increased GMV in the bilateral hippocampus, left superior temporal gyrus, left middle temporal gyrus, left superior occipital gyrus and other brain areas. More importantly, social and communication behaviors are related to the gray matter structure of specific brain regions, indicating that the occurrence of clinical symptoms in ASD children is related to changes in their gray matter structure. ASD is characterized by impairment of social function. Among the brain regions mentioned above, the abnormality of the hippocampal structure and function is closely related to the social deficiency of ASD [27]. This study found that the hippocampal GMV of ASD significantly increased, which is consistent with previous research results [28]. In ASD patients, structural changes also often exist in the subcortex composed of gray matter nuclei under the cerebral cortex. The VBM analysis showed that the GMV of ASD patients decreased, mainly in the striatum, amygdala, hippocampus and parahippocampal gyrus [29].

Changes in the hippocampus may lead to decreased episodic memory, spatial reasoning, and social interaction abilities [30,31,32]. Social problems of ASD mainly include difficulty in facial recognition, such as eye avoidance, low accuracy of facial recognition, and long reaction time for facial recognition [33,34], so this will lead ASD to deficiencies in emotional understanding, emotional expression, emotional regulation, and social skills [35]. Episodic memory deficits have been reported in ASD, and recent functional magnetic resonance imaging studies have found that it may be caused by the decreased connectivity between the hippocampus and other brain regions [36,37,38]. Other studies have shown a connection between episodic memory and cognitive impairment in ASD [39]. In addition, compared to TD individuals, ASD patients have difficulties in memorizing faces and depicting scenes of human interaction, indicating that memory is particularly impaired under social stimuli, which can affect social behaviors and may lead to problems with peer interaction.

In this study, a regression analysis further found that GMV in specific brain regions is associated with clinical symptoms in the ASD group, which may help explain the potential brain mechanism of social and communication problems with ASD. Specifically, we found a positive correlation between the subtotal score of communications and social interaction on ADOS and the left hippocampus GMV. Hippocampus is a key part of emotional regulation in the nervous system and is considered the pathophysiological basis of ASD. Groen [40] found that the left hippocampus GMV of ASD patients is significantly larger than that of TD individuals matched with age, sex and IQ, which is consistent with the results of this study. An abnormal increase in hippocampus volume in adolescents with ASD may change their emotional understanding and emotional regulation because the components of the limbic system mediate memory, social and emotional functions, which are usually interfered with in ASD. The development defect of the limbic system has been confirmed as the basis of ASD symptoms [41]. Previous findings indicate that abnormal patterns of hippocampal development in ASD persist throughout childhood and adolescence [42]. The hippocampal volume of ASD children was very small at first, and it increased abnormally with age. The hippocampal volume in ASD children was negatively correlated with ASD’s nonverbal intelligence [43]. Some studies have shown that the volume of the right hippocampus of ASD patients is larger than that of normal adults, and the dysplasia pattern of the hippocampus in ASD patients will continue to adolescence [44].

However, Barnea-Goraly [45] studied the development track of the hippocampus in children with ASD aged 8 to 14 years, and the correlation between the volume change and clinical symptoms was examined. It was found that there was no correlation between the change in hippocampus volume in the ASD group and the score of social interaction in ADI-R, but the right hippocampus volume in children with ASD increased significantly. The right hippocampus volume in ASD patients decreased significantly over time, but hippocampus volume in the healthy control group was stable or slightly increased. No significant correlation was found between hippocampus volume and social interaction score, which may be closely related to the number, gender and age of participants.

This study found that ASD children showed increased GMV in the left superior temporal gyrus and left middle temporal gyrus compared with the TD children, which was consistent with previous research results. Cai [46] compared 19 low-functional ASD children and 19 high-functional ASD children with 27 healthy controls and found that GMV in the left inferior temporal gyrus of ASD children was increased compared with that of healthy controls, but only GMV in the left middle temporal gyrus of high-functional ASD children was increased. Similar findings have been found in ASD adults, where VBM exhibited changes in brain anatomical structure with clinical phenotype, and ASD patients showed an increased GMV in the frontal and temporal lobes [47]. A study found no significant difference in GMV in the right superior temporal gyrus between the ASD group and the TD group in cohorts aged 13–18 and 19–30, which also supports the results of this study [3]. ASD is a highly heterogeneous disease which has different manifestations in language development, intelligence and comorbidity [48,49]; It is often observed in defects in the early language development track in ASD children [50]. One study found that temporal pole GMV was significantly enlarged in ASD children with developmental delay and is associated with language ability during the pre-language stage and the severity of social impairment in ASD children with developmental delay [51]. This seems to indicate that an increase in temporal pole GMV may provide a biological basis for the severity of autism before the onset of clinical symptoms. The increase in temporal pole GMV indirectly supports the results of this study.

In addition, this study found that the GMV of the left middle temporal gyrus and superior temporal gyrus in the ASD group was positively correlated with the subtotal score of communications and social interaction. Consistent with the results of this study, it has been previously found that GMV in the temporal and frontal lobes of ASD patients increases, and these structural abnormalities are positively correlated with ADI-R scores [52]. In addition, a previous structural connectivity study showed that the disconnection of the temporal and frontal lobes was positively correlated with communication dysfunction [53]. The Wernicke area near the superior temporal gyrus of the cerebral cortex is a classic language processing area [54,55]. It is related to the symptoms of language comprehension and expression disorders in ASD patients [56].

Research has found that language stimulus paradigms activate functional language regions such as the temporal lobe [57]. This is crucial for word understanding, semantics, naming, reading, and spelling, and the key areas related to these functions include the left superior temporal gyrus [58]. Meanwhile, the lesions of the superior temporal gyrus, middle temporal gyrus, inferior temporal gyrus, middle frontal gyrus, inferior frontal gyrus, and inferior frontal gyrus will lead to hearing impairment [59]. Previous studies have found that the temporal lobe is closely related to language function [60,61] and social function [62,63]. The superior temporal gyrus is the core cortical area of the social brain, which has the function of integrating sensory information and limbic system information [64,65,66]. A meta-analysis based on task state MRI research found that in social tasks, the activation level of the right superior temporal gyrus in ASD children was lower than that in ASD adults [67], indicating that structural abnormalities in these regions may be a potential cause of social communication problems in ASD children.

A previous meta-analysis has revealed a widespread reduction of GMV when comparing ASD with TD [68]. The reasons for inconsistency with previous research results are likely due to different image processing methods [69], unknown sex differences in brain morphology [70], different collecting approaches and limited sample size with heterogeneous characteristics of subjects [71]. Interestingly, in addition to the cerebrum, studies have found that a reduction of GMV in different cerebellar subregions is associated with increased social communication problems and repetitive behavior [72]. This may be related to the different functional roles of the regions in the cerebellum in motor, cognitive, and emotional processing [73] because the cerebellum can receive input from the associative regions of the prefrontal and parietal lobes and participate in cognitive tasks [74]. Future research directions can focus on structural changes in the cerebellar subregions.

A longitudinal study found that the GMV of ASD children increased in early childhood, and it was close to the level of the control group until late childhood. In addition, the study also found that the larger the corpus callosum GMV of ASD is, the lower the ADOS score will be [75]. Longitudinal research can help elucidate the developmental neuropathology of ASD throughout their lifecycle and suggest the necessity of continuing to examine brain structures throughout the entire lifecycle and adulthood. Cross-sectional studies can infer brain changes from age-related differences between individuals. Because of the fact that longitudinal research typically takes several years and requires collecting data from multiple time points and tracking participants, it is difficult to conduct longitudinal research. However, longitudinal research can directly measure changes within individuals and more accurately describe changes in brain development, maturation, and aging [76]. Longitudinal studies can not only compare brain development changes at different periods but also make logical causal judgments more easily than horizontal studies due to the clear time order of various variables [77]. The correlation analysis between abnormal brain regions and the severity of social communication problems in our study has further proved the relationship between changes in GMV in some specific brain regions and clinical symptoms. The findings in our study are partly consistent with previous research results. The abnormal volume of GMV in the left superior temporal gyrus, left middle temporal gyrus and left hippocampus probably provides support for the social brain hypothesis of ASD, which is helpful in understanding the neuroanatomy of ASD. In conclusion, this study leveraged the ABIDE database to investigate brain structure differences between ASD children and TD children. Significant differences in total brain volume determined by IQ have been reported in previous studies [78]. Therefore, after considering intelligence factors, the difference between the two groups of children found in this study is more convincing. However, there are several limitations to this study. Firstly, this study is a cross-sectional design, and future longitudinal studies are needed to address the dynamic abnormalities of brain structure in ASD patients; secondly, this study is limited by the ABIDE database, and factors such as height, weight, and parenting style of ASD patients still need to be considered. Thirdly, the social brain network is a brain network responsible for social functions, composed of multiple different brain regions involved in social cognitive processes. Therefore, future research should pay attention to the changes in social brain structure during the development of ASD, and it is necessary to further explore the changes in brain network level and the relationship between structural covariation and social communication problems in ASD through the Structural Covariance Network.

## 5. Conclusions

In summary, this study investigated the GMV of ASD children after controlling for factors such as gender, age, total intracranial volume, and IQ. It was found that compared to TD children, the GMV of multiple brain structures in ASD children increased. In addition, social and communication dysfunction in ASD children is associated with structural abnormalities in specific brain structures. These findings contribute to understanding the potential brain mechanisms of ASD children and may provide evidence to explain the clinical symptoms of ASD.

## Figures and Tables

**Figure 1 brainsci-13-00779-f001:**
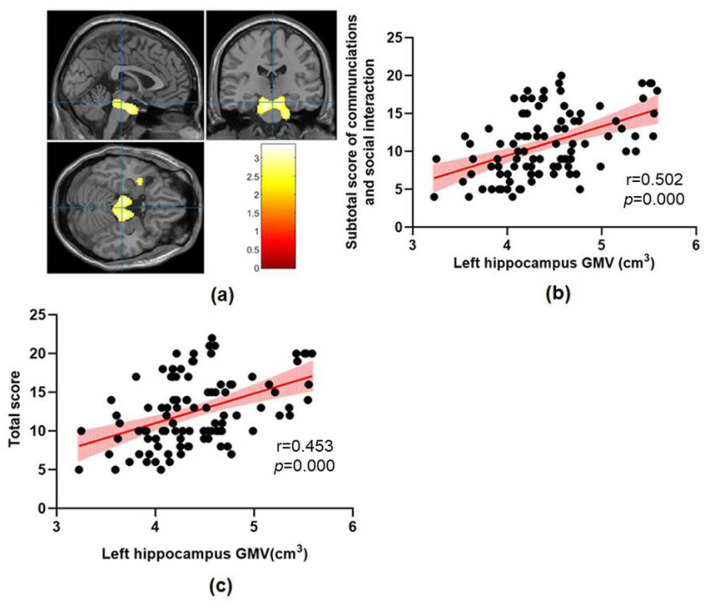
(**a**) The increased GMV in ASD children compared to TD children in cluster 1. (**b**) The correlation between left hippocampus GMV and subtotal score of communications and social interaction on ADOS. (**c**) The correlation between left hippocampus GMV and total score on ADOS.

**Figure 2 brainsci-13-00779-f002:**
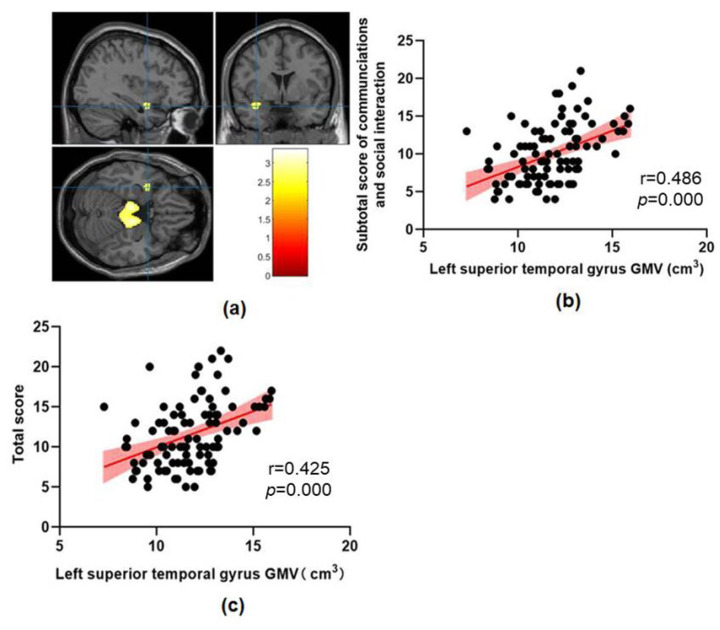
(**a**) The increased GMV in ASD children compared to TD children in cluster 2. (**b**) The correlation between left superior temporal gyrus GMV and subtotal score of communications and social interaction on ADOS. (**c**) The correlation between left superior temporal gyrus GMV and total score on ADOS.

**Figure 3 brainsci-13-00779-f003:**
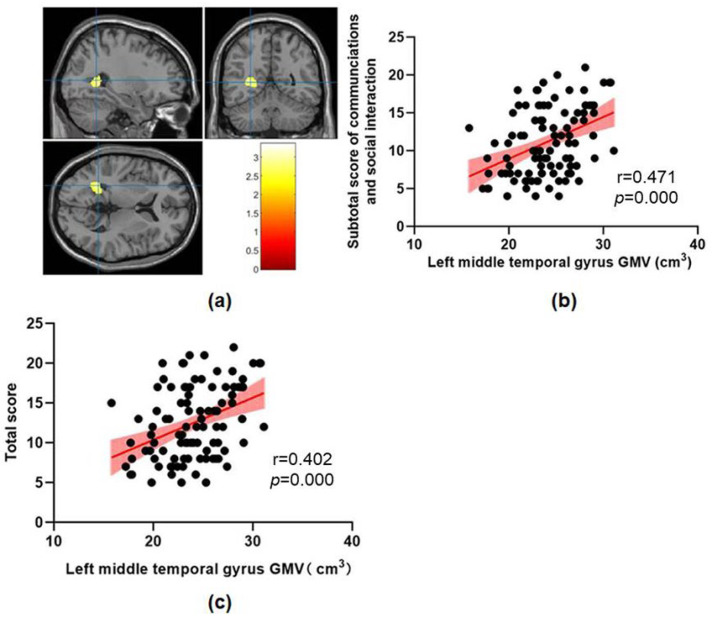
(**a**) The increased GMV in ASD children compared to TD children in cluster 3. (**b**) The correlation between left middle temporal gyrus GMV and subtotal score of communications and social interaction on ADOS. (**c**) The correlation between left middle temporal gyrus GMV and total score on ADOS.

**Table 1 brainsci-13-00779-t001:** Demographic information.

	ASD (n = 98) Mean ± SD	TD (n = 105) Mean ± SD	t/χ^2^	*p*
Age	10.17 ± 1.23	10.06 ± 1.34	0.54	0.49
Gender (male/ female)	67/31	64/41	4.48	0.06
Full IQ	111.75 ± 15.32	113.82 ± 12.58	−1.46	0.38
ADOS scores				
Communication score		3.54 ± 1.45		
Social score		7.75 ± 2.68		
Stereotypic behavior score		2.41 ± 1.65		
Subtotal score of communications and social interaction Total score		11.29 + 3.82 13.70 ± 4.21		

**Table 2 brainsci-13-00779-t002:** Regions showing significant differences in gray matter volume between ASD and TD groups based on VBM analyses.

Clusters	Peak MNI			Voxel Number	Regions	t	*p*
	X	Y	Z				
ASD > TD							
Cluster1	−9	−18	−25.5	5299	Midbrain, Pontine Bilateral hippocampus, Left parahippocampal gyrus	3.28	0.001
Cluster2	−39	3	−19.5	170	Left superior temporal gyrus, Left temporal pole	2.57	0.005
Cluster3	−31.5	−52.5	6	544	Left middle temporal gyrus, Left superior occipital gyrus	2.88	0.002
ASD < TD	No significant results						

Note: MNI: Montreal Neurological Institute.

## Data Availability

Data are available on request due to restrictions of privacy or ethics; the data presented in this study are available on request from the corresponding author. The data are not publicly available due to privacy.

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
