# Peer review of "Abnormal Brain Structure Is Associated with Social and Communication Deficits in Children with Autism Spectrum Disorder: A Voxel-Based Morphometry Analysis"

_brainsci, 2023, doi:10.3390/brainsci13050779_

Round 1
Reviewer 1 Report
The present study uses ABIDE data set to look for the correlation between volumetric brain data and core symptoms of ASD. However, the topic, introduction, and discussions are not supported by the results as the statistical analysis does not include a correlation with "score for communications + social interaction". Relevant literature is not cited including arousing papers of ABIDE. Please go through previous that used ABIDE data sets to appreciate how they have characterized the data set, acknowledged ABIDE, ect.
Major comment
Comment 1
The latest edition of ADOS is ADOS-2, which version have you used?
ADOS assesses several domains of ASD including Communication, Social interaction, Imagination and creativity, and Repetitive behaviors. (https://doi.org/10.1177%2F1362361317698938)
However, the title, introduction, etc implies that the present study focuses on Social and Communication defects.
e.g., Title; Abnormal Brain Structure is Associated with “Social and Communication Deficits”
Line 213-215; In addition, this study found that the GMV of left middle temporal gyrus and superior temporal gyrus in ASD group was positively correlated with the “total score of social interaction and communication”.
However, in the results, the GMV is correlated with the total ADOS score, not the subtotal of “communications + social interaction”. This makes the paper scientifically weak.
If you intend to do a re-analysis you should also perform a correlation of GMV with “communications + social interaction” and revise table 1 with the addition of a row for “communications + social interaction”
Comment 2
Line 10-11:; but the relationship between structural changes and social communication disorders is still unclear.
Line 62-63; relationship between structural changes and social communication disorders is still unclear.
There is a separate entity in DSM-5 called “social communication disorder (SCD)”. SCD may be a distinct diagnosis or may co-occur with other conditions. But, in the case of autism spectrum disorder, social communication problems are a defining feature, along with restricted, repetitive patterns of behavior. Therefore, SCD cannot be diagnosed in conjunction with an autism spectrum disorder.
Reference; https://www.psychiatry.org/File%20Library/Psychiatrists/Practice/DSM/APA_DSM-5-Social-Communication-Disorder.pdf
I believe what you are focusing on in this study is the relationship between structural changes and “social communication problems”/social communication “impairment”/social communication “deficits”. Therefore, please avoid the term “social communication disorder”
Comment 3
Volumetric brain studies with correlations to ADOS has been reported before but not cited the present paper. Consult the following
Ji Y, Xu M, Liu X, Dai Y, Zhou L, Li F, Zhang L. Temporopolar volumes are associated with the severity of social impairment and language development in children with autism spectrum disorder with developmental delay. Front Psychiatry. 2022 Dec 1;13:1072272. doi: 10.3389/fpsyt.2022.1072272. PMID: 36532174; PMCID: PMC9751401.
Duan Y, Zhao W, Luo C, Liu X, Jiang H, Tang Y, Liu C, Yao D. Identifying and Predicting Autism Spectrum Disorder Based on Multi-Site Structural MRI With Machine Learning. Front Hum Neurosci. 2022 Feb 22;15:765517. doi: 10.3389/fnhum.2021.765517. PMID: 35273484; PMCID: PMC8902595.
D'Mello AM, Crocetti D, Mostofsky SH, Stoodley CJ. Cerebellar gray matter and lobular volumes correlate with core autism symptoms. Neuroimage Clin. 2015 Feb 20;7:631-9. doi: 10.1016/j.nicl.2015.02.007. PMID: 25844317; PMCID: PMC4375648.
Riva D, Annunziata S, Contarino V, Erbetta A, Aquino D, Bulgheroni S. Gray matter reduction in the vermis and CRUS-II is associated with social and interaction deficits in low-functioning children with autistic spectrum disorders: a VBM-DARTEL Study. Cerebellum. 2013 Oct;12(5):676-85. doi: 10.1007/s12311-013-0469-8. PMID: 23572290.
Additional volumetric brain studies have been reported before but not cited the present paper.
Prigge MBD, Lange N, Bigler ED, King JB, Dean DC 3rd, Adluru N, Alexander AL, Lainhart JE, Zielinski BA. A 16-year study of longitudinal volumetric brain development in males with autism. Neuroimage. 2021 Aug 1;236:118067. doi: 10.1016/j.neuroimage.2021.118067. Epub 2021 Apr 18. PMID: 33878377; PMCID: PMC8489006.
Comment 4
If you use data from the ABIDE Preprocessed repository, need to cite their paper:
The autism brain imaging data exchange: towards a large-scale evaluation of the intrinsic brain architecture in autism. Mol Psychiatry. 2014 Jun;19(6):659-67.
doi: 10.1038/mp.2013.78. Epub 2013 Jun 18.
Comment 5
Acknowledgment:
As per the Usage Agreement http://fcon_1000.projects.nitrc.org/indi/abide/abide_I.html , the specific datasets included in analyses should be characterized and specified appropriately, and their funding sources should be acknowledged.
For example
Consult the following
Nielsen JA, Zielinski BA, Fletcher PT, Alexander AL, Lange N, Bigler ED, Lainhart JE, Anderson JS. Multisite functional connectivity MRI classification of autism: ABIDE results. Front Hum Neurosci. 2013 Sep 25;7:599. doi: 10.3389/fnhum.2013.00599. PMID: 24093016; PMCID: PMC3782703.
Minor comments
Comment 6
Line 53-55; “Problem behaviors of ASD such as unusually destructive behaviors, are related to the damage of its amygdala and related temporal lobe structures.”
Is this an established finding? The paper you have cited is just a hypothesis, hence use terms like “possible relationship”, and “difficult to establish” in the abstract.
Please see the paraphrased sentence.
Unusually destructive behaviors in ASD may be related to damage to the amygdala and related temporal lobe structures.
Comment 7
Line 58-59 “However, inconsistent conclusions have been drawn from the study that no abnormalities in temporal GMV were found in ASD patients when gender and age were controlled [20] ”
Reference [20] is an earlier study, not one of the two studies [18.19] you cited earlier. So instead of “the study” use “an earlier study” to prevent confusion. The conclusions are contradictory and but we are not so sure which study contributed to inconsistency. I would suggest the following modifications to the sentence so that the message is clearer.
“However, contradictory conclusions have been drawn from an earlier that no abnormalities in temporal GMV were found in ASD patients when gender and age were controlled”
Comment 8
Line 72-73 and 78-79; ABIDE has been defined twice. Can be defined only at first mention. Please check other abbreviations as well (e.g. ADOS,WASI).
satisfactory
Reviewer 2 Report
The sentences are circuitous, The findings are not clearly described.
The authors should more concisely described main finding on the abnormal brain structures ant their association to the subscales of ADOS.
Reviewer 3 Report
Manuscript is very competently prepared. I have no further edits to suggest.
Round 2
Reviewer 1 Report
Thanks for sending the revised versions back to me. I am delighted that the authors have agreed to my comments and fixed all of them. I would accept the version as it is. However, I would be grateful if you can convey this message to the authors. Comment 1 I feel authors can still keep the correlations with the total score (in addition to the subtotal they have analyzed) as it's statistically significant, and readers may be interested to know. The authors have deleted the total score from Table 1 and the figures. Comment 2 "However, contradictory conclusions have been drawn from an earlier" > However, contradictory conclusions have been drawn from an earlier study..."
Reviewer 2 Report
The association between abnormal brain structure and the bscores of the ADOS subscale scires was described in children with ASD, The findings may be excellent.
